# Analysis and Regulation of the Harmonious Relationship among Water, Energy, and Food in Nine Provinces along the Yellow River

**Jiawei Li [1], Junxia Ma [1,2,*], Lei Yu [1,2,3] and Qiting Zuo [1,2,3]**

1 School of Water Conservancy Engineering, Zhengzhou University, Zhengzhou 450001, China; lijiawei6869@163.com (J.L.); yulei2018@zzu.edu.cn (L.Y.); zuoqt@zzu.edu.cn (Q.Z.)
2 Henan International Joint Laboratory of Water Cycle Simulation and Environmental Protection, Zhengzhou 450001, China
3 Yellow River Institute for Ecological Protection & Regional Coordinated Development, Zhengzhou University, Zhengzhou 450001, China
* Correspondence: majx@zzu.edu.cn

**Abstract:** China has proposed "ecological conservation and high-quality development of the Yellow River Basin" to a major national strategy, which puts forward higher requirements for water, energy, and food along the Yellow River (TYR). However, the water–energy–food nexus (WEF) system in TYR basin is very complicated. Based on the theory and method of harmonious regulation, this paper puts forward a new WEF harmony framework (WEFH) to study the harmonious balance of WEF in TYR. WEFH cannot only evaluate the harmonious balance of WEF, but also identify the main influencing factors, and further study the harmonious regulation of WEF. For the key steps of regulation and control, we provide a variety of methods to choose from in this framework. In practice, we apply this framework to the regulation of WEF in the nine provinces along TYR. The results show that during 2005–2018, the harmony degree of WEF in the nine provinces along TYR is between 0.29 and 0.58. The harmony degree of WEF has improved over time, but there is still a lot of room for improvement. Among them, per capita water resources, hydropower generation ratio, carbon emissions, and another 12 indicators have great influence on the harmony of WEF. We have established eight control schemes for nine of these indicators. In eight control schemes, most areas have reached a moderate level of harmony degree. These results show that the framework proposed in this paper is helpful to the comprehensive management of regional WEF and provides a viable scheme for the optimization of WEF.

**Keywords:** water–energy–food; harmony equilibrium; harmonious regulation; the Yellow River

## 1. Introduction

Water, energy, and food are important strategic resources, which are closely interlinked with each other. They are important building blocks for economic and social development and national security [1]. Since the three are interdependent, changes in any area may alter their supporting or constraining roles and upset the balance among them. Therefore, effective research on the coordinated development of water, energy, and food is fundamental in order to promote high-quality regional development [2].

In 2011, the Global Risks 2011 Report (6th edition) suggested that there are complex relationships among WEF and their risks are one of the three most important global risks. Since then, scholars have carried out a series of studies on WEF. The studies related to WEF initially started with the water resources subsystem and gradually expanded from single subsystem studies to integrated studies of two and three subsystems. Therefore, the research for WEF includes three main categories. On individual subsystem studies,

there are numerous studies that address water, energy, and food, respectively. Water subsystem research involves water resources [3–5], water rights [6,7], water environment [8], water energy [9,10], etc. Energy subsystem research involves resources [11,12], carbon emissions [13–15], energy optimization [16], etc. Food subsystem research involves food security [17], planting optimization [18,19], etc. The two subsystem studies include water–energy [20], water–food [21], etc. The integrated study of the three subsystems involves the concept of WEF [22], the relationship of WEF [23], and the optimization of WEF [24,25], etc.

As China's "Mother river", the Yellow River (TYR) basin is an important "Energy basin" and "Agricultural basin" [26]. There are important energy bases and grain-growing areas distributed in TYR basin, which produce one third of grain and meat output in China [27]. With the ecological protection and high-quality development of TYR basin becoming a national strategy, in the reality of limited water resources in TYR basin, how to solve the problems of the water system has attracted more attention. According to existing studies, TYR has special spatial and temporal distribution characteristics of water, energy, and food, which has a profound impact on the regional harmony of WEF [28]. The concept of "harmony" is derived from the "harmonious society" proposed by China. Later the Chinese Ministry of Water Resources proposed the 'Human–Water Harmony', which embodies China's beautiful wish for a harmonious coexistence between humans and nature. Energy and food, as important aspects involved in human systems, are studied in harmony with water in order to reflect the state of local water use, energy exploitation, and food security. Therefore, it is of great significance to analyze the current situation of TYR and study the harmonious degree of WEF, which is a basal content, in order to realize the strategy of ecological protection and high-quality development of TYR.

Based on the above analysis, this paper intends to investigate the spatial and temporal evolution and harmony regulation of WEF in the nine provinces along TYR. Section 2 gives an overview of the study area. Section 3 introduces the methods and data that are used in this paper. Section 4 presents the results and discussion, which analyzes the spatial and temporal evolution characteristics of three subsystems, harmony level evaluation results of WEF, harmony identification results, and harmonious regulation results. Section 5 provides the conclusion and the outlook for the future [29–31].

## 2. Study Area

After the Yangtze River, TYR is the second longest river in China. Its mainstream is 5464 km, ranking fifth in the world. The total area of the river basin is 795,000 km², accounting for about 8.3% of the total land surface area of China [32]. The basin involves nine provinces and 62 major cities along the river. TYR originates in Qinghai, flowing in turn through nine provinces, which include Gansu, Sichuan, Ningxia, Inner Mongolia, Shaanxi, Shanxi, Henan, and Shandong. The geographic location of TYR and an overview of the WEF are shown in Figure 1.

The WEF of TYR basin is extremely complex due to its own properties and complex human activities along TYR. TYR basin is relatively scarce in water resources, with an exploitation rate as high as 80%, the average annual runoff is only 7% of the Yangtze River basin, and the per capita water resources are only 905 m³, far below the national average. However, TYR basin involves 8% of the population and 9% of GDP in China. Along the route, there are a number of major food production areas and rich mineral resources [33]. A series of typical features, such as high population density, wide distribution of industry and agriculture, and intensive human activities in TYR basin, make its WEF complex and significant for research.

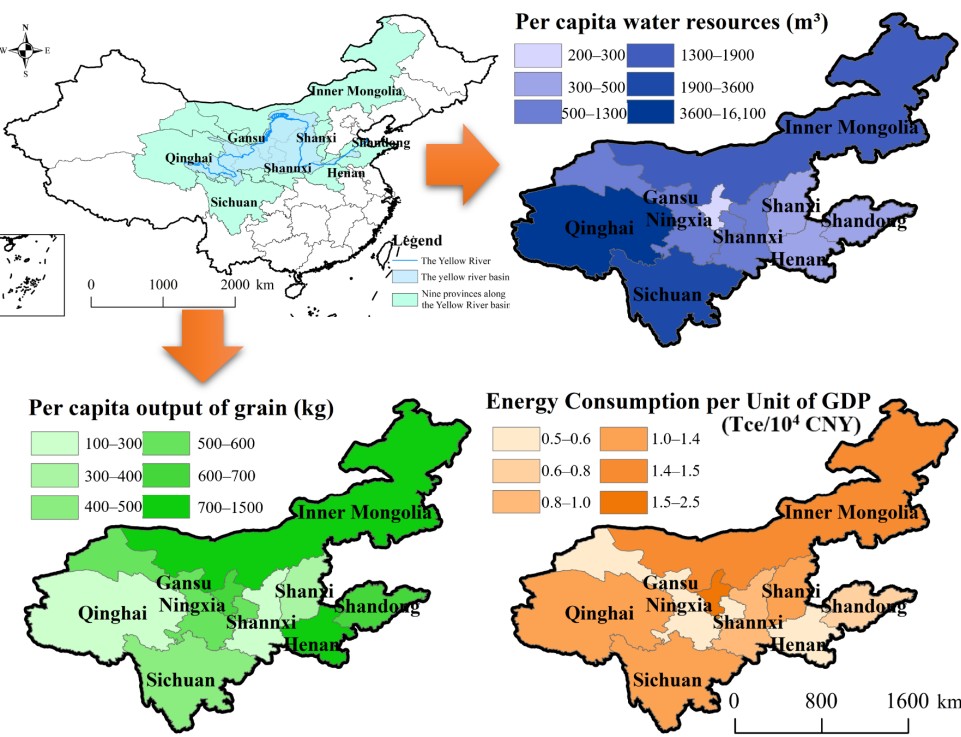

**Figure 1.** Overview map and water, energy, and food distribution of TYR (2018).

## 3. Methods and Data

### 3.1. Research Ideas and Framework

This paper puts forward a new water–food–energy harmony framework (WEFH) to study the harmonious balance of WEF in TYR. Starting from the spatial–temporal evolution, WEFH has sequentially conducted harmony evaluation and regulation studies. As shown in Figure 2, it consists of the following four main steps:

Step 1: Present situation and problems. The spatial and temporal evolution characteristics of the water, energy, and food subsystems are analyzed by selecting representative indicators for each subsystem. Next, the current problems are summarized. This is the basis and urgent need for the harmony assessment.

Step 2: Harmony assessment. Evaluate the harmony degree of each subsystem and WEF. It includes the following three parts: indicator system construction, weight calculation, and comprehensive evaluation. Among them, principal component analysis (PCA) [34] can be used to construct the indicator system, analytic hierarchy process (AHP) [35] can be used for weight determination, and single-indicator quantification, multi-indicator synthesis, and multi-criteria integration (SMI-P) [36,37] can be used for comprehensive evaluation. WEFH is an open framework, and other methods can be added according to the actual situation.

Step 3: Harmony identification. Identify the main influencing factors and screen the indicators with greater influence [37]. This is the premise and foundation of harmonious regulation. WEFH provides a variety of identification methods for reference. In this paper, the obstacle degree model is used [29].

Step 4: Harmonious regulation. Based on the assessment of harmony, harmony regulation improves the degree of harmony by taking some regulatory measures to make the participants of harmony develop in the direction of harmony [37]. In this paper, we simulate the harmonious regulation of WEF through scenario design [29].

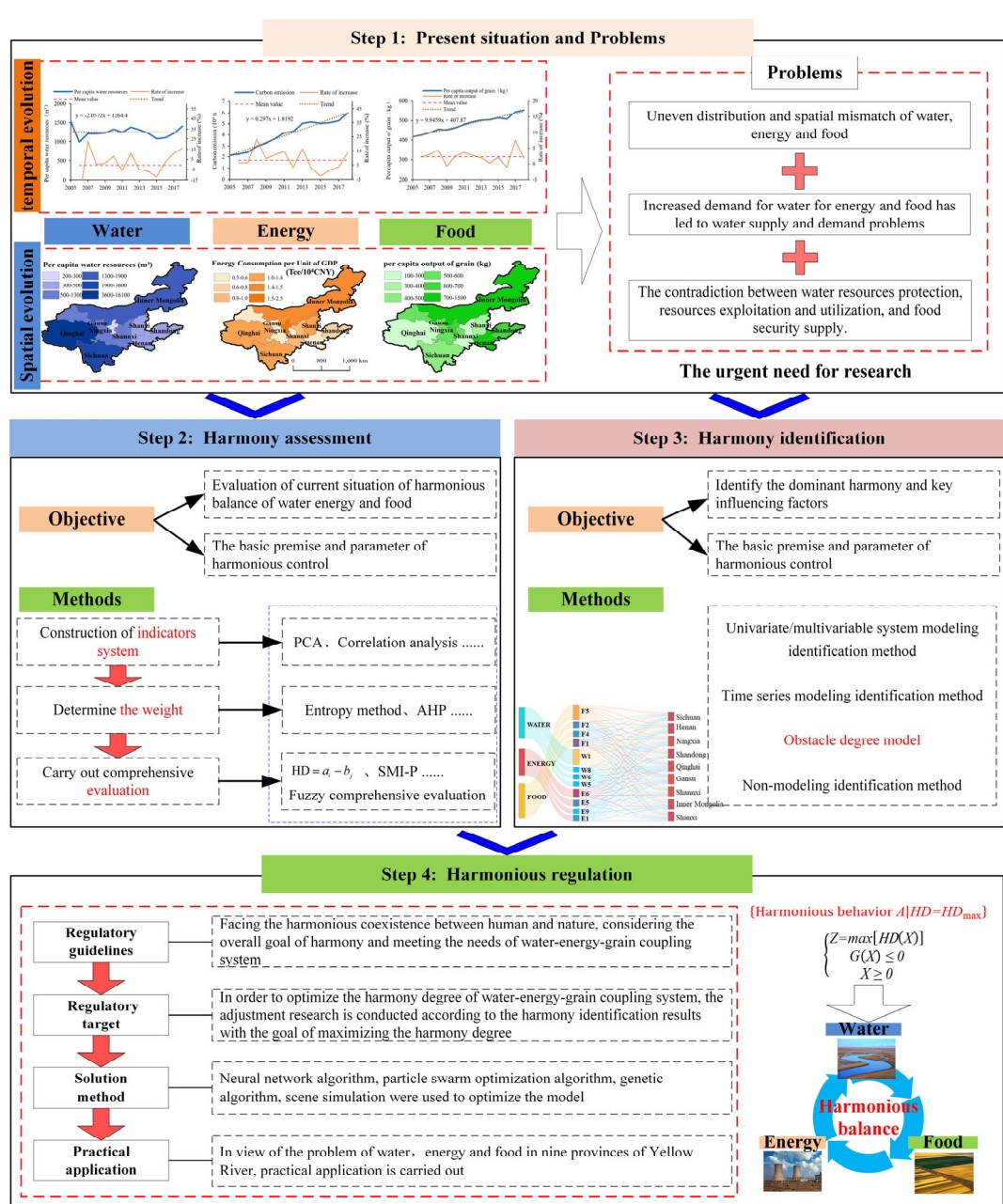

**Figure 2.** Framework for analysis and regulation of the harmonious relationship among water, energy, and food (water–energy–food harmonious, WEFH).

### 3.2. Spatial–Temporal Evolution Analysis Method

The WEF system consists of the following three subsystems: water, energy, and food. Based on water flow and energy flow, it includes a series of processes, such as constraints, feedback, and adaptation, between and within each subsystem. Based on the understanding of the WEF system, this paper selects the main elements from three subsystems, water, energy, and food, for spatial and temporal evolution analysis.

Taking into account the actual conditions of TYR basin, the availability of data, and the depth of research, this paper analyzes the spatial and temporal evolution characteristics of per capita water resources, carbon emissions, and per capita food production for the three subsystems. Temporally, the linear tendency estimation method is used to analyze the temporal evolution characteristics and calculate the linear trend of the selected elements. Spatially, the spatial distribution characteristics of the selected elements were analyzed by using ArcGIS.

Linear tendency estimation method: The tendency rate (*Slope*) of the time series data is calculated to characterize the changing trend of the data over time. The calculation formula is as follows:

$$Slope = \frac{n\sum_{i=1}^{n}(iK_i) - \sum_{i=1}^{n}i\sum_{i=1}^{n}K_i}{n\sum_{i=1}^{n}i^2 - \left(\sum_{i=1}^{n}i\right)^2} \tag{1}$$

where *Slope* is the tendency rate. If *slope* > 0, it indicates that the system element shows an increasing trend. If *slope* < 0, the opposite is true; $n$ is the length of the sample sequence; $K_i$ is the statistical data of the $i$ year.

### 3.3. Harmony Assessment Method

According to the second step of WEFH, this paper first selects 39 candidate indicators to represent the level of WEF harmony. Secondly, PCA is used to eliminate some indicators with multicollinearity and small contribution rate in order to determine the final indicator system of the nine provinces along TYR. Third, the subjective weight method (AHP) and the objective weight method (the entropy weight method) are used to determine the weights of each indicator. Finally, this paper uses the SMI-P [36] method to evaluate the harmony degree of WEF [37].

### 3.3.1. Construction of Indicator System

WEF as a comprehensive system, each of its subsystems covers complex indicators. Considering the characteristics of WEF, the actual situation and data availability of each subsystem, 13, 11, and 15 indicators are selected for the three subsystems of water, energy, and food, respectively. The generic indicator system is shown in Table 1. In the application, we use PCA to filter the indicators in Table 1 [38,39]. Only those indicators that are relevant will be retained.

**Table 1.** Candidate indicator system for WEF harmony assessment.

| Target | Subsystem | Indicators | Unit | Attribute |
|---|---|---|---|---|
| WEF's harmonious balance | WATER | Per capita water resources | $m^3$/per head | + |
| | | Per capita water consumption | $m^3$/per head | − |
| | | Proportion of industrial water consumption | % | − |
| | | Proportion of groundwater supply | % | − |
| | | Reclaimed water reuse rate | % | + |
| | | Total wastewater discharge | $10^4$ t | − |
| | | Discharge of chemical oxygen demand (COD) in wastewater | $10^4$ t | − |
| | | Proportion of ecological water consumption | % | + |
| | | Water penetration rate | % | + |
| | | Average daily wastewater treatment capacity | $10^4$ $m^3$/d | + |
| | | Length of drainage pipeline | km | + |
| | | Length of water supply pipeline | km | + |
| | | Comprehensive production capacity of water supply | $10^4$ $m^3$/d | + |
| | ENERGY | Energy consumption per unit of GDP | Tce/$10^4$ CNY | − |
| | | Electricity consumption | $10^8$ kW·h | − |
| | | Power generation | $10^8$ kW·h | + |
| | | Primary energy output (equivalent value) | $10^4$ tce | + |
| | | Investment in energy industry | $10^8$ CNY | + |
| | | Proportion of hydropower generation | % | + |
| | | Added value of the secondary industry | $10^8$ CNY | + |
| | | Natural gas production | $10^4$ $m^3$ | + |
| | | Coal base reserves | $10^8$ t | + |

| | Carbon emission | t | – |
|---|---|---|---|
| | Production of general industrial solid waste | $10^4$ t | – |
| | Gross agricultural output | $10^8$ CNY | + |
| | Gross output value of agriculture, forestry, animal husbandry, and fishery | $10^8$ CNY | + |
| | Per capita output of grain | kg/per head | + |
| | Per capita output of pig, beef, and mutton | kg/per head | + |
| | Arable land | $10^4$ hm² | + |
| | Effective irrigation area | $10^3$ hm² | + |
| FOOD | Grain sown area | $10^3$ hm² | + |
| | Agricultural land area | $10^4$ hm² | + |
| | Total power of agricultural machinery | $10^4$ kW | + |
| | Agricultural fertilizer yield | $10^4$ t | – |
| | Irrigation water consumption per unit area | m³ | – |
| | Per capita grain consumption of rural households | kg | – |
| | Area affected by the disaster | $10^3$ hm² | – |
| | Urban Engel coefficient | % | – |
| | Rural Engel coefficient | % | – |

According to the PCA method, this paper eliminates the variables with high correlation and repeated connotation from the selected candidate indicators. On the basis of ensuring the integrity of the indicator information, some variables are selected as the final indicator [38,39]. The steps of PCA are as follows:

a. Assuming that there are $m$ years of data, and each year has $n$ quantitative indicators, an $m \times n$ matrix $A$ is obtained as follows:

$$A = \begin{vmatrix} x_{11} & x_{12} & \cdots & x_{1n} \\ x_{21} & x_{22} & \cdots & x_{2n} \\ \vdots & \vdots & & \vdots \\ x_{m1} & x_{m2} & \cdots & x_{mn} \end{vmatrix} \tag{2}$$

where $x_{mn}$ is the indicator data;

b. Standardize matrix $A$ to obtain matrix $B$ as follows:

$$b_{ij} = \frac{\left(x_{ij} - \bar{x}_j\right)}{s_j} \tag{3}$$

Where $b_{ij}$ is an element of matrix $B$, $s_j$ is the standard deviation;

c. Calculate the correlation coefficient matrix $C$ of the standardized matrix $B$, and then calculate the $n$ eigenvalues of $C$ and the unit eigenvector of the eigenvalues;

d. Sort according to the size of the eigenvalues, and calculate the contribution rate $a_j$ of the principal components;

e. Calculate the principal component coefficient matrix $D$ and arrange the coefficients from largest to smallest. It reflects the correlation between the indicator and the principal component;

f. Calculate the correlation coefficients for the indicators. When the correlation coefficient is greater than 0.8, we consider the indicators to be highly correlated, and need to be deleted as redundant information.

Because the variance of the principal components can reflect indicators with larger component coefficients, the indicators with larger component coefficients in each principal component are retained, and the indicators with multicollinearity and low principal component contribution rate are eliminated.

3.3.2. Weight Determination

In order to scientifically measure the weights, considering the pros and cons of subjective weights and objective weights, this paper combines the analytic hierarchy process (AHP) and entropy weight method to determine the weights of each indicator based on the least square method [40].

(1) Analytic hierarchy process (AHP)

AHP is based on the experience of decision makers to determine the relative importance of indicators in the overall system. Divided into the following 2 basic steps:

a.　Construct the judgment matrix, as follows:

Construct a judgment matrix $A = (a_{ij})_{n \times n}$. For a certain element in the upper layer, compare the importance of each element in the next layer pair by pair. $A_{ij}$ uses a 9-bit scaling method to take the value, which can be 1, 2... 9 and its reciprocal.

b.　Calculate the weight vector and eigenvalue, as follows:

Determine the weight vector $W = (w_1, w_2 \ldots w_n)^T$ and eigenvalues according to the judgment matrix, as follows:

$$\mathrm{w}_i = \frac{1}{n} \sum_{j=1}^{n} \frac{a_{ij}}{\sum_{k=1}^{n} a_{kj}}, i = 1, 2, \cdots, n \tag{4}$$

$$\lambda = \frac{1}{n} \sum_{i=1}^{n} \frac{\sum_{j=1}^{n} a_{ij} w_j}{w_i} \tag{5}$$

(2) Entropy method

The entropy weighting method is based on the variability of indicators to calculate the objective weights. For the evaluation indicator matrix $X = (x_{ij})_{m \times n}$ with $m$ evaluation indicators and $n$ evaluation objects, the calculation steps are as follows [41]:

Step 1: Normalized processing is performed as follows:

a.　For positive indicators:

$$y_{ij} = \frac{x_{ij} - \min\{x_{ij}\}}{\max\{x_{ij}\} - \min\{x_{ij}\}} \tag{6}$$

b.　For contrarian indicators:

$$y_{ij} = \frac{\max\{x_{ij}\} - x_{ij}}{\max\{x_{ij}\} - \min\{x_{ij}\}} \tag{7}$$

Step 2: Determine the entropy weight $w_i$ as follows:

$$H_i = \frac{-\sum_{j=1}^{n} f_{ij} \ln f_{ij}}{\ln n} \tag{8}$$

$$f_{ij} = y_{ij} / \sum_{j=1}^{n} y_{ij} \tag{9}$$

$$w_i = \frac{1 - H_i}{m - \sum_{i=1}^{m} H_i} \tag{10}$$

(3) Combined weight based on least square method

In order to realize the unification of the subjective and objective weight calculation methods in the indicator weighting, the combined weight model based on the least square method is used to determine the combined weight [40]. The formula is as follows:

$$A = \mathrm{diag} \left[ \sum_{i=1}^{n} z_{i1}^2, \sum_{i=1}^{n} z_{i2}^2, \cdots, \sum_{i=1}^{n} z_{im}^2 \right] \tag{11}$$

$$B = \left[ \sum_{i=1}^{n} \frac{1}{2}(u_1 + v_1) z_{i1}^2, \sum_{i=1}^{n} \frac{1}{2}(u_2 + v_2) z_{i2}^2, \cdots, \sum_{i=1}^{n} \frac{1}{2}(u_m + v_m) z_{im}^2 \right]^T \tag{12}$$

$$W = A^{-1} \cdot \left[ B + \frac{1 - e^T A^{-1} B}{e^T A^{-1} e} \cdot e \right] \tag{13}$$

where $A$ is the diagonal array, and $W$ and $B$ are vectors.

### 3.3.3. Harmony Evaluation

This paper adopts the method of "single-indicator quantification, multi-indicator synthesis, multi-criteria integration (SMI-P)" to evaluate the harmonious degree of WEF in the nine provinces along TYR [37]. Among them, single-indicator quantification quantifies each indicator by fuzzy affiliation function, and maps the indicators to [0, 1] interval by setting 5 node values to eliminate the influence of dimensionality and positive and negative indicators; multi-indicator synthesis is achieved by weighting the affiliation degree of each indicator to achieve a comprehensive study of multiple indicators; multi-criteria integration is calculated by weighting each subsystem to produce a final composite index [30,31].

### 3.4. Harmony Identification Method

WEF involves a large number of influencing factors, and there are differences in the magnitude of the role of different influencing factors on the level of harmony. In order to identify the main factors affecting the level of regional WEF harmony, this paper uses the obstacle degree model to diagnose the obstacle factors and identify the main influencing factors. The calculation steps of the obstacle degree model are as follows [37]:

a.    Calculate the factor contribution $F_j$ of evaluation indicator $j$ as follows:

$$F_j = w_j w_j^* \tag{14}$$

where $w_j^*$ is the weight of the criterion layer to which indicator $j$ belongs.

b.    Calculate the deviation degree $I_j$ as follows:

$$I_j = 1 - x_{ij} \tag{15}$$

c.    Calculate the obstacle degree $P_j$ of each evaluation indicator as follows:

$$P_j = \frac{F_j I_j}{\sum_{j=1}^{n} F_j I_j} \tag{16}$$

### 3.5. Harmonious Regulation Method

Based on the results of the WEF harmony assessment and with reference to the main influencing factors obtained from harmony identification, harmony regulation research is conducted on WEF in the nine provinces along TYR. Harmony regulation is performed by taking some regulation measures to improve the degree of harmony based on the harmony assessment, so that the harmony participants will develop in the direction of harmony. It mainly includes two ideas [37], as follows:

(1)   Harmonious behavior set preference method: Harmonious solutions are determined by comparing the magnitude of the harmony of each solution in the behavior set as follows:

$$HD_{max=max}\{HD_k\} \ (k = 1, 2, 3 \cdots n) \tag{17}$$

where $HD_k$ is the harmony degree of the $k$ scheme.

(2)   Based on the optimization model of harmony degree, through the adjustment model, the adjustment measures that meet the requirements are calculated as follows:

$$\begin{cases} Z = max[HD(X)] \\ \quad G(X) \leq 0 \\ \quad\quad X \geq 0 \end{cases} \tag{18}$$

where $Z$ is the objective function value, $X$ is the decision vector, $HD(X)$ is the objective function, and $G(X)$ is the set of constraints.

### 3.6. Data Source

The time scale of data used in this paper is 2005–2018, and various statistics are obtained from China Statistical Yearbook, China Energy Statistical Yearbook, and Water Resources Bulletin, etc.

## 4. Results and Discussion

### 4.1. Characteristics of Temporal and Spatial Evolution

#### 4.1.1. Evolution Characteristics of Water Subsystem Elements

The evolution characteristics of per capita water resources are shown in Figure 3. The southwest part is mainly mountainous with better vegetation and more abundant water resources. The northern region has a dry climate, low annual precipitation, and poorer water resources. The per capita water resources in the central region are placed between the two, but soil erosion is serious. The per capita water resources in the nine provinces show a fluctuating decreasing trend with an average value of 1248 m³. Among the nine provinces, Qinghai's per capita water resources are much higher than those of the other regions.

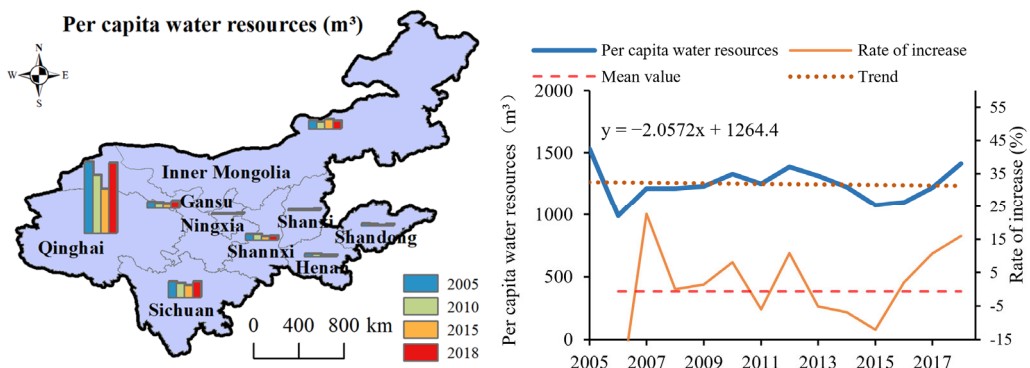

**Figure 3.** Distribution and trend of per capita water resources.

#### 4.1.2. Evolution Characteristics of Energy Subsystem Elements

The wind and solar energy resources in the upper reaches of TYR and the coal and oil and gas resources in the middle and lower reaches are important resources to support China's economic development. As an important energy base in China, TYR basin has a high proportion of coal production and consumption, and the development and utilization of fossil energy have brought great pressure on the ecological environment and water resources utilization, and the task of low-carbon emission reduction is heavy. Carbon emissions in TYR basin have been increasing year by year since 2005. The evolution characteristics of carbon emissions are shown in Figure 4. Under the demand of "high-quality development", the energy production and consumption structure of TYR basin needs to be transformed and upgraded.

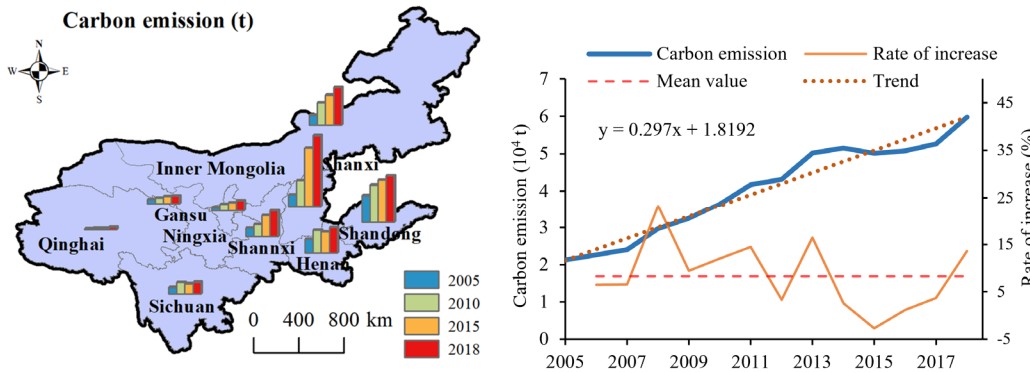

**Figure 4.** Distribution and trend of carbon emission.

### 4.1.3. Evolution Characteristics of Food Subsystem Elements

TYR basin is a key region to ensure food security in China. Food security has always been one of the major issues of great concern to China. In 2018, the nine provinces and regions in TYR basin produced 232,688,700 t of grain. The evolution characteristics of per capita grain production are shown in Figure 5. The per capita output of grain in the nine provinces show a fluctuating growth trend. This indicates a significant increase in the region's food production capacity. Among them, Sichuan, Inner Mongolia, Henan, and Shandong are the main grain producing areas in the country.

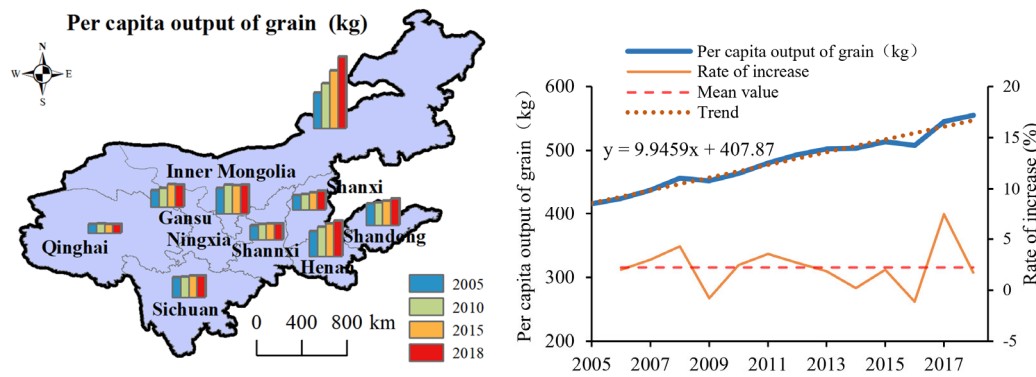

**Figure 5.** Distribution and trend of per capita grain output.

### 4.2. Harmony Level Evaluation Results

#### 4.2.1. Indicator System Screening and Node Values

Based on the candidate indicator system in the previous section (Table 2), PCA is conducted for the current situation of TYR basin. The results of PCA are combined with qualitative analysis in order to determine the final indicator system, as shown in Table 2.

**Table 2.** WEF harmony evaluation indicator system in the nine provinces along TYR.

| Target | Subsystem | Indicators | Number |
|--------|-----------|------------|--------|
| WEF's harmonious balance | WATER | Per capita water resources | W1 |
| | | Per capita water consumption | W2 |
| | | Proportion of industrial water consumption | W3 |
| | | Proportion of groundwater supply | W4 |
| | | Reclaimed water reuse rate | W5 |
| | | Total wastewater discharge | W6 |
| | | Discharge of chemical oxygen demand (COD) in wastewater | W7 |

| | | |
|---|---|---|
| | Daily sewage treatment capacity | W8 |
| | Length of drainage pipe | W9 |
| | Comprehensive production capacity of water supply | W10 |
| | Energy consumption per unit of GDP | E1 |
| | Electricity consumption (physical volume) | E2 |
| | Power generation | E3 |
| | Primary energy output (equivalent value) | E4 |
| ENERGY | Investment in energy industry | E5 |
| | Proportion of hydropower generation | E6 |
| | Natural gas production | E7 |
| | Coal base reserves | E8 |
| | Carbon emissions | E9 |
| | Production of general industrial solid waste | E10 |
| | Gross agricultural output | F1 |
| | Per capita food output | F2 |
| | Arable land | F3 |
| | Effective irrigation area | F4 |
| FOOD | Total power of agricultural machinery | F5 |
| | Agricultural fertilizer yield | F6 |
| | Irrigation water consumption per unit area | F7 |
| | Inundated area | F8 |
| | Urban Engel coefficient | F9 |
| | Rural Engel coefficient | F10 |

After determining the indicator data, we determine the node values of each indicator. Combined with the real situation of TYR basin and the indicator properties, five nodal values were divided for each indicator as follows: best, better, pass, worse, and worst. The nodal values are divided by the multi-year average value of each indicator in each region as the qualified value; the highest value is expanded by 10% as the optimal value, where the percentage of indicators reaching 100% is not expanded; the lowest value is reduced by 10% as the worst value; the worse value and the better value are determined by the interpolation method, and the nodal characteristic values and indicator weights of the indicators are shown in Table 3.

**Table 3.** Quantified indicator node feature values and indicator weights.

| Number | Best | Better | Pass | Worse | Worst | AHP | Entropy Weight Method | Combination Weight |
|---|---|---|---|---|---|---|---|---|
| W1 | 17,794.59 | 10,073.42 | 2352.25 | 1237.10 | 121.96 | 0.33 | 0.28 | 0.30 |
| W2 | 147.76 | 296.36 | 444.95 | 942.86 | 1440.77 | 0.24 | 0.04 | 0.14 |
| W3 | 3.98 | 9.58 | 15.18 | 22.88 | 30.59 | 0.04 | 0.06 | 0.05 |
| W4 | 3.58 | 18.06 | 32.55 | 51.28 | 70.02 | 0.17 | 0.08 | 0.12 |
| W5 | 6.71 | 4.04 | 1.38 | 0.69 | 0.01 | 0.03 | 0.09 | 0.06 |
| W6 | 17,424.00 | 96,313.55 | 175,203.09 | 408,411.28 | 641,619.47 | 0.12 | 0.05 | 0.08 |
| W7 | 5.18 | 31.63 | 58.09 | 138.08 | 218.08 | 0.02 | 0.05 | 0.03 |
| W8 | 1345.36 | 821.75 | 298.15 | 152.90 | 7.65 | 0.02 | 0.11 | 0.07 |
| W9 | 70,586.40 | 41,259.04 | 11,931.68 | 6215.32 | 498.97 | 0.02 | 0.14 | 0.08 |
| W10 | 2081.33 | 1342.22 | 603.10 | 333.86 | 64.61 | 0.03 | 0.10 | 0.06 |
| E1 | 0.53 | 1.03 | 1.52 | 3.04 | 4.56 | 0.32 | 0.05 | 0.18 |
| E2 | 185.90 | 902.43 | 1618.96 | 4155.68 | 6692.40 | 0.02 | 0.03 | 0.02 |

| | | | | | | | | |
|---|---|---|---|---|---|---|---|---|
| E3 | 6408.17 | 4150.06 | 1891.94 | 1043.12 | 194.30 | 0.02 | 0.05 | 0.04 |
| E4 | 90,643.30 | 56,814.64 | 22,985.98 | 12,444.22 | 1902.47 | 0.04 | 0.11 | 0.07 |
| E5 | 3721.30 | 2405.92 | 1090.55 | 583.95 | 77.36 | 0.17 | 0.05 | 0.11 |
| E6 | 96.12 | 58.57 | 21.02 | 10.51 | 0.01 | 0.13 | 0.16 | 0.14 |
| E7 | 24,999.70 | 13,305.66 | 1611.62 | 805.81 | 0.00 | 0.01 | 0.24 | 0.13 |
| E8 | 1167.66 | 696.40 | 225.13 | 116.83 | 8.52 | 0.03 | 0.17 | 0.10 |
| E9 | 19.01 | 234.32 | 449.62 | 1205.49 | 1961.35 | 0.27 | 0.11 | 0.19 |
| E10 | 584.10 | 6271.14 | 11,958.18 | 26,614.54 | 41,270.89 | 0.01 | 0.03 | 0.02 |
| F1 | 5471.05 | 3519.82 | 1568.60 | 800.70 | 32.80 | 0.03 | 0.14 | 0.08 |
| F2 | 1543.99 | 1006.53 | 469.07 | 307.37 | 145.66 | 0.33 | 0.12 | 0.22 |
| F3 | 1019.92 | 751.00 | 482.09 | 265.44 | 48.80 | 0.03 | 0.08 | 0.06 |
| F4 | 5817.56 | 4032.03 | 2246.51 | 1202.60 | 158.69 | 0.04 | 0.12 | 0.08 |
| F5 | 14,688.32 | 9309.57 | 3930.81 | 2112.71 | 294.61 | 0.12 | 0.18 | 0.15 |
| F6 | 22.62 | 166.30 | 309.98 | 695.44 | 1080.90 | 0.02 | 0.15 | 0.08 |
| F7 | 131.41 | 259.97 | 388.53 | 819.93 | 1251.34 | 0.24 | 0.07 | 0.15 |
| F8 | 38.07 | 345.22 | 652.36 | 1739.85 | 2827.33 | 0.02 | 0.05 | 0.03 |
| F9 | 20.46 | 26.58 | 32.70 | 40.55 | 48.40 | 0.02 | 0.05 | 0.03 |
| F10 | 22.77 | 29.50 | 36.22 | 47.52 | 58.82 | 0.17 | 0.04 | 0.10 |

### 4.2.2. Evaluation Results of Each Subsystem

(1) Water subsystem

The results of the water subsystem harmony assessment in the nine provinces of TYR are shown in Figure 6. The overall trend of water subsystem harmony in the nine provinces and regions is increasing, and the harmony range is [0.31, 0.66]. The worst value appeared in Shanxi province in 2005, and the best value appeared in Sichuan province, which is closely related to the local water resources endowment and water use patterns. Among them, Henan, Ningxia, and Shanxi initially had lower water subsystem harmony, but with increased ecological awareness and technological progress, Shanxi and Henan have improved their water system harmony levels, which is of reference to Ningxia and other regions. Sichuan and Qinghai have a better foundation of water system harmony, but it is not obvious with the growth of time, for example, Sichuan is in a higher state of water subsystem harmony from the beginning to the end.

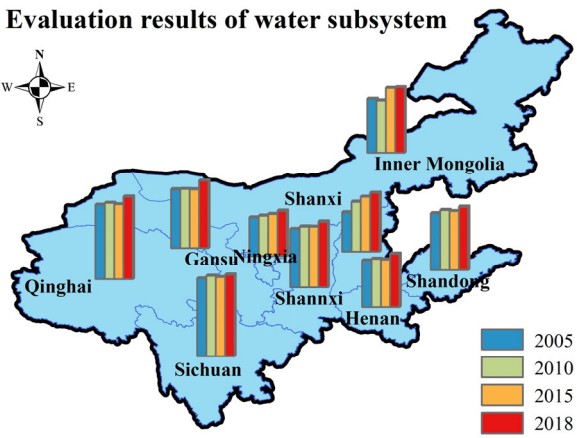

**Figure 6.** Evaluation results of the water subsystem in the nine provinces of TYR.

(2) Energy subsystem

The results of the energy subsystem harmony assessment in the nine provinces are shown in Figure 7. The results show that the overall energy subsystem harmony in the nine provinces and regions shows an increasing trend, with a harmony range of [0.26,

0.57]. The worst value appears in Ningxia in 2005, and the best value appears in Gansu province, which is mainly influenced by the local natural resource endowment and energy consumption. Among them, Ningxia, Qinghai, Sichuan, and Henan initially had low energy subsystem harmony, but with industrial progress and improvement of production methods, Sichuan and Ningxia achieved some improvement in energy system harmony level; Henan, Inner Mongolia, and Shanxi provinces showed fluctuating changes or even a decline in energy system harmony level; Gansu has a better foundation of energy system harmony and has progressed over time. It is in the leading position among the regions.

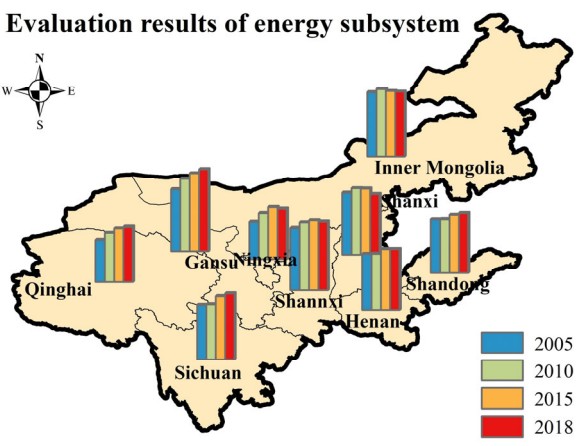

**Figure 7.** Assessment results of energy subsystems in the nine provinces of TYR.

(3) Food subsystem

The results of the food subsystem harmony assessment in the nine provinces are shown in Figure 8. The results show that the overall trend of food subsystem harmony in the nine provinces and regions is increasing, and the harmony range is [0.20, 0.81]. The worst value appears in Qinghai in 2005, and the best value appears in Henan and Shandong, which is closely related to the local grain production and agricultural level. Among them, Qinghai, Gansu, and Ningxia initially had lower food subsystem harmony, but with technological progress and the improvement of production methods, the level of food system harmony in Gansu and Ningxia has been improving, which is of reference to Qinghai and other regions. Henan and Shandong have a better foundation of food system harmony and have progressed rapidly over time, confirming that Henan and Shandong are famous grain producing areas.

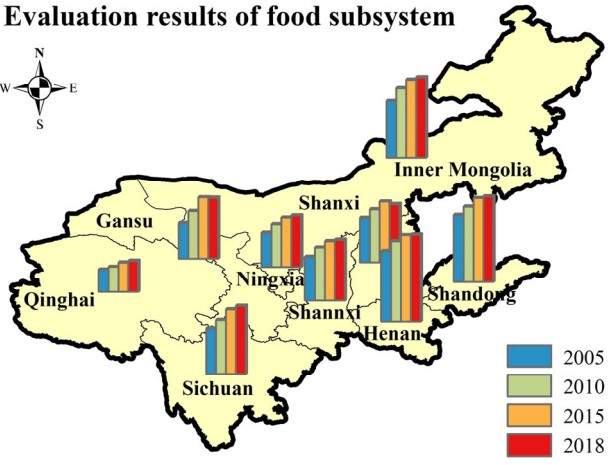

**Figure 8.** Evaluation results of the food subsystem in the nine provinces of TYR.

### 4.2.3. Evaluation Results of WEF Harmony

Overall, the WEF harmony degree in the nine provinces ranges from 0.29 to 0.58, as shown in Figure 9. On the time scale, all nine of the provinces and regions show a year-by-year growth trend, and the WEF harmony degree keeps improving. This is closely related to the year-by-year growth of water harmony, food harmony, and energy optimization in each region. On the spatial scale, Ningxia and Qinghai have a lower WEF harmony degree than the other regions, which is closely related to the region's poor natural resource endowment and rash water use, energy consumption, and food consumption. Among them, Ningxia is at a lower level in all three of the subsystems, and Qinghai has better water endowment but lower energy and food harmony levels, which together lead to a lower overall harmony level.

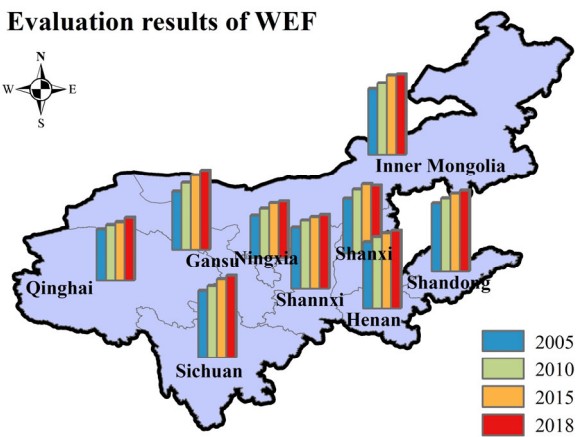

**Figure 9.** WEF harmony assessment results of the 9 provinces of TYR.

### 4.3. *Analysis of Harmony Identification Results*

The obstacle degree of each province was calculated, and the indicators with higher obstacle degrees were selected as the main obstacle factors affecting the level of WEF harmony. The 12 indicators with higher obstacle degrees are shown in Figure 10. They include four water system indicators, four energy system indicators, and four food indicators. In the harmonized regulation, the indicators with a higher degree of impairment are regulated.

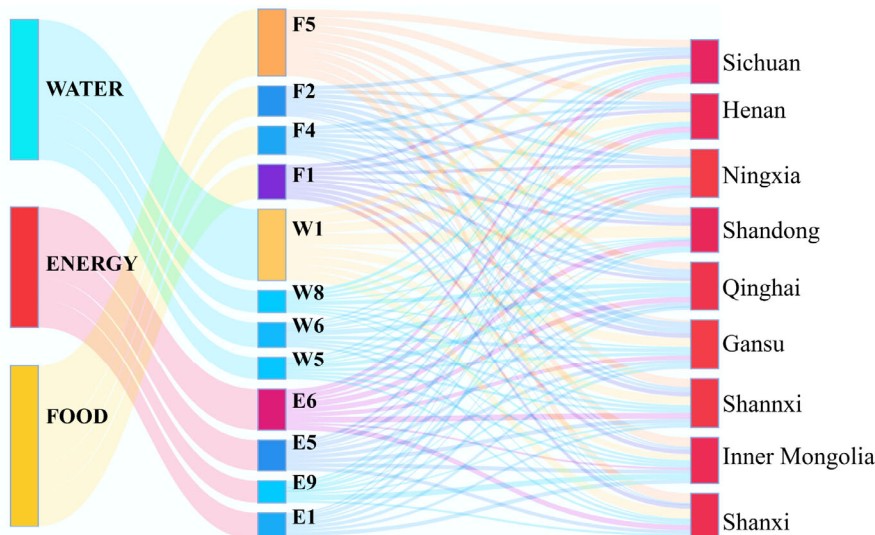

**Figure 10.** Key obstacle indicators and subsystems influencing the harmony degree of WEF in nine provinces.

According to the obstacle degree model. There are 12 main indicators affecting the harmony balance level in the nine provinces and regions of TYR. Among them, per capita water resources (W1), natural gas production (E7), and per capita grain production (F2) have the greatest influence on the harmony level of each region. There are minor differences among regions, but they are mainly influenced by these 12 factors.

### 4.4. Analysis of Harmonious Regulation Results

Based on the results of the WEF harmony assessment in the nine provinces of TYR, the harmonious balance of the nine provinces of TYR is at a moderate to low level, and there is still much room for improvement. Therefore, this paper conducts a harmonious regulation study on WEF in the nine provinces with reference to the main influence factors obtained from the harmonious identification. The harmonious behavior set preference method is used to calculate the regulation measures that meet the requirements.

For the 12 indicators with higher obstacle degrees, some cannot be subjectively regulated artificially due to their natural properties, such as natural gas production and coal reserves. Considering the adjustability of the indicators and the actual scope of regulation, as well as the spatial and temporal evolution characteristics of each subsystem element, the regulation study is conducted on the basis of 2018. After optimizing 2% of the basic indicators, two regulation schemes of high (H) and medium (L) are set for the key impact factors obtained from the harmonious identification, and a total of eight schemes are formed, as shown in Table 4.

**Table 4.** Harmonious regulation scheme setting.

| | Control Indicators | Harmonious Regulation Plan (H = 10%, L = 5%) | | | | | | | |
|---|---|---|---|---|---|---|---|---|---|
| | | Scheme 1 | Scheme 2 | Scheme 3 | Scheme 4 | Scheme 5 | Scheme 6 | Scheme 7 | Scheme 8 |
| WATER | Per capita water resources | H | H | H | H | L | L | L | L |
| | Daily sewage treatment capacity | H | H | L | L | H | H | L | L |
| | Recycle rate of wastewater | H | L | H | L | H | L | H | L |
| ENERGY | Hydropower generation ratio | H | H | H | H | L | L | L | L |
| | Carbon emission | H | H | L | L | H | H | L | L |
| | Energy consumption per unit of GDP | H | L | H | L | H | L | H | L |
| FOOD | Total power of agricultural machinery | H | H | H | H | L | L | L | L |
| | Per capita output of grain | H | H | L | L | H | H | L | L |
| | Effective irrigation area | H | L | H | L | H | L | H | L |

We set up eight schemes according to Table 4 and calculated the harmony degree of WEF under each scheme. The results are shown in Table 5. After harmonious regulation, the degree of harmony has been significantly improved compared to the original level, and most areas have reached a medium level of harmony.

**Table 5.** Harmony degree of different harmony control schemes.

| Province | Scheme 1 | Scheme 2 | Scheme 3 | Scheme 4 | Scheme 5 | Scheme 6 | Scheme 7 | Scheme 8 | 2018 |
|---|---|---|---|---|---|---|---|---|---|
| Gansu | 0.59 | 0.58 | 0.58 | 0.57 | 0.58 | 0.58 | 0.58 | 0.57 | 0.56 |
| Henan | 0.58 | 0.58 | 0.58 | 0.57 | 0.58 | 0.57 | 0.57 | 0.57 | 0.55 |
| Inner Mongolia | 0.60 | 0.60 | 0.60 | 0.60 | 0.60 | 0.59 | 0.59 | 0.59 | 0.57 |
| Ningxia | 0.41 | 0.41 | 0.41 | 0.41 | 0.41 | 0.41 | 0.41 | 0.41 | 0.40 |
| Qinghai | 0.46 | 0.46 | 0.46 | 0.46 | 0.46 | 0.46 | 0.46 | 0.46 | 0.44 |
| Shandong | 0.60 | 0.59 | 0.59 | 0.59 | 0.59 | 0.59 | 0.59 | 0.59 | 0.57 |
| Shanxi | 0.50 | 0.50 | 0.49 | 0.49 | 0.49 | 0.49 | 0.49 | 0.49 | 0.47 |
| Shannxi | 0.55 | 0.55 | 0.55 | 0.54 | 0.54 | 0.54 | 0.54 | 0.54 | 0.52 |
| Sichuan | 0.61 | 0.61 | 0.61 | 0.61 | 0.61 | 0.61 | 0.60 | 0.60 | 0.58 |

## 5. Conclusions

In this paper, a harmonious evaluation index system was constructed with WEF as the research object. The evolutionary characteristics of the representative elements of each of the subsystems were analyzed. The harmony degree of the nine provinces along TYR was studied and the harmony regulation of WEF was carried out. This paper draws several conclusions, as follows:

(a)  The representative elements of the subsystem have different distribution characteristics. The per capita water resources of TYR were 1248.98 m$^3$. It shows the distribution characteristics were high in the west and low in the east. The carbon emissions were much higher in the east than in the west. Among them, Shanxi and Shandong had larger carbon emissions. The per capita output of grain is increasing. Among them, Inner Mongolia, Henan, and Shandong had larger per capita grain production. Based on this result, each province can identify its own strengths and weaknesses. This is very useful for the provinces to maintain their strengths and make up for their shortcomings;

(b)  In this paper, 30 indicators were selected in order to evaluate the harmonious relationship of WEF in the nine provinces along TYR. The evaluation results of the water subsystem show a gradual increase and the distribution was higher in the west and lower in the east. However, the energy and food subsystems were higher in the east. WEF were not fully aligned spatially. The results of the WEF show that the harmony degree of WEF in the nine provinces ranged from 0.29 to 0.58, which is at a medium level. Among them, Ningxia and Qinghai are worse, while Sichuan, Shandong, and Inner Mongolia are better. There is some room for regulation;

(c)  The main indicators influencing the harmonious balance of the WEF were calculated based on the obstacle degree model. The per capita water resources (W1), natural gas production (E7), and per capita grain production (F2) have a strong influence on the level of harmony. These indicators point the way to harmonious regulation and serve as a reference for individual provinces;

(d)  This paper sets up eight scenario simulation scenarios and calculates the harmony of WEF under each scenario. After the harmony regulation, most of the provinces along TYR reach the medium level. The study can provide a reference for the regulation of each region. Different provinces can regulate the WEF in response to their own problems.

Due to the complex and variable relationships of WEF, the harmonious analysis of WEF in this paper is superficial and macroscopic. Facing the needs of high-quality development of TYR, there are still some shortcomings in this paper. (a) The analysis of temporal and spatial evolution of the subsystems is inadequate. Considering the research focus, this paper selects only one representative element for each subsystem in the practical application. It can be systematically studied in further research. (b) Whether the indicator system can entirely represent the relationship of WEF needs to be further explored. This is a problem that all of the indicator systems need to face, and the representativeness of the indicator system for different regions or countries needs to be analyzed according to the actual national context. (c) Lack of consideration of inter-provincial transfer of resources, which should be deepened in future studies. (d) There are some shortcomings in the schemes setting and these schemes only provide some guidance. There is a lack of specific schemes.

**Author Contributions:** Conceptualization, J.M.; methodology and investigation, J.L. and J.M.; formal analysis, L.Y.; writing—original draft preparation, J.L.; writing—review, editing, and supervision, Q.Z. and L.Y.; project administration and funding acquisition, Q.Z. All authors have read and agreed to the published version of the manuscript.

**Funding:** This research was funded by the National Key Research and Development Program of China (No. 2021YFC3200201) and the Major Science and Technology Projects for Public Welfare of Henan Province (No. 201300311500).

**Institutional Review Board Statement:** Not applicable.

**Informed Consent Statement:** Not applicable.

**Acknowledgments:** The authors are grateful to the editors and the anonymous reviewers for their insightful comments and suggestions.

**Conflicts of Interest:** The authors declare no conflict of interest.

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
