# Peer review of "Analysis and Regulation of the Harmonious Relationship among Water, Energy, and Food in Nine Provinces along the Yellow River"

_water, doi:10.3390/w14071042_

Round 1
Reviewer 1 Report
I am positively surprised by the quality of the submitted paper! Without being ground-breaking research, the authors manage to convey their message clearly, in a well written paper, joyful to read, without unnecessary text and yet with all the crucial information provided.
I really do not have much to say here beyond that I would expect some more discussion on the “whys” (especially not being an expert with the Chinese context of water-food-energy nexus). Those why would refer equally to the different evolutionary trajectories of the regions as well as the actual implications in terms of management. Yet this goes beyond the scope of the paper quite clearly introduced by the authors in the introduction.
In short: the authors deliver successfully what they promise…Not rocket-science for sure, but definitely interesting and relevant enough for the audience of the journal.
Actually I only have a minor comment concerning the conclusions: the authors nicely identify the weaknesses of the paper. Yet they do so in a very “dry” was if I may say, without providing some further hints on the direction of future research. For example, although they state that “this paper selects only one representative element for each subsystem in the practical application. It can be systematically studied in further research” (lines 416-417) they do not elaborate further on the representativeness of this one representative element and how a future analysis can be different in a systematic way. Similarly, in lines 417-419 they argue “Whether the indicator system can adequately represent the relationship of WEF needs to be further explored”: well if it doesn’t represent the relationship then the whole analysis looses its value? I am not sure that this last sentence was clear and meaningful enough. Lastly the four concluding points could be elaborated slightly more (a coupel of sentences on actual implications and value of the findings).
All in all, I think the paper is almost ready for publication, yet I would highly recommend a short enhancement of the concluding section.
Author Response
Dear Reviewer,
Thank you very much for your comments and suggestions on our manuscript entitled " Analysis and Regulation of the Harmonious Relationship among Water, Energy and Food in nine provinces along the Yellow River " (ID: Water-1615475). Those comments and suggestions are very valuable and helpful for revising and improving our manuscript.
The manuscript has been deeply revised according to reviewers’ comments and suggestions, which are highlighted by using the "Track Changes" function in Microsoft Word.
The corrections in the manuscript and responses to you are listed below. We would like to re-submit this revised manuscript to WATER, and hope this manuscript is suitable for the journal now.
We deeply appreciate your hard work and consideration of our manuscript, and look forward to hearing from you soon.
Thank you and best regards.

Reviewer 2 Report
The subject is current and very important. The goal of this research was investigate the spatial and temporal evolution and harmony regulation of water-energy-food (WEF) in 9 provinces along the Yellow River. The authors developed the harmonious evaluation index system with WEF. The evolutionary characteristics of the representative elements of each subsystems was analyzed. The harmony degree of 9 provinces along the Yellow River was studied and the harmony regulation of WEF was carried out. My basic remarks to the paper:
• In the Conclusions, the authors directly point to the shortcomings of the method used. I really like such a critical approach of the authors themselves to a very broad and complicated problem. I consider the work as an interesting introduction to further research and deepening the subject.
• The results of the research studies were well analyzed.
• It is a pity that data older than 2005 were not taken into account.
The submitted paper made a good impression on me. Therefore, I have no strong remarks. However, moderate changes to the English language are required. The manuscript is well structured and deserves publication after some minor revisions.
Author Response

(The authors gave the same response as above.)

Reviewer 3 Report
In this well-expressed paper, the authors have sought to ambitiously apply the principles of developing “harmonious balance” across nine provinces of the Yellow River Basin for the joint complexities of water, energy and food production. The principle of the harmonious degree of Water, Energy and Food, and its regulation is introduced in lines 62-66 of page 2 of the draft manuscript. However, the philosophy is not explained or defined, and much of the published harmony literature seems to have been oriented to one parameter, such as water, a somewhat less complex system.. What is the ultimate objective being sought? How does this relate to the principles of “sustainability”? Though subject to some development by, principally, Chinese authors, the harmony concepts are little known elsewhere. While the Methods section explains the process, it is essential that the authors define what they are discussing (perhaps then quoting reference 38) before embarking on describing the research undertaken. Note that some parts of Figure 2 are not legible on screen and certainly not when printed.
The area of the Yellow River basin is extremely large and with great variability of resources. It is noted that some resources have been expressed on a per capita basis – for example W1 (water resources) is a function of geographical location, not a function of the population. Therefore it is not surprising that Qinghai, which is mountainous with high rainfall and low population has a high value for W1. It is noted that Inner Mongolia has a seemingly rapid increase in the food system results. But is this a function of genetic improvement in crop varieties from a low base or a function of the migration of part of the population to cities in other provinces? Similarly, although the paper explores provincial resources at a rather macro-level, it does not acknowledge the impact of water use behaviours at smaller sub-basin level, an example being the overuse of surface water for irrigation in high country west of the North China Plain and the consequent loss of natural groundwater replenishment on the Plain itself.
The paper assumes a principle of the self-sufficiency within each Province. It does not appear to provide for inter-provincial transfer of resources, yet the Yellow River itself is an agent of inter-provincial resource transfer There seem to be some inconsistencies in the selection of indicators. Water resources are listed in various forms including reclaimed water, ground water, groundwater discharge and could be assumed to be essential for production and environmental systems. On the other hand, coal based reserves are noted in some provinces, but ultimately could become irrelevant as to energy economy moves to renewable energy sources to meet China’s commitment to a low-emissions future. The wind and solar resources of the upper reaches of TYR Basin are noted, but do not appear to have been included in the energy indicators and modelling, yet one would expect them to be increasingly important in the future and their energy productivity likely to be widely exported through inter-provincial transfers.
The authors have generally shown an improvement in harmony evaluation results in the various provincial sub-systems (water, energy, food), but could devote more discussion to hypothesising how this has been achieved and also to the issues to be faced in future such as the impact of climate change on water resources and the consequences of decisions in response to emissions reduction policies. It is unclear whether the modelling assumes a status quo steady environmental condition then influenced by provincial anthropogenic impacts within that system, or whether it also acknowledges the environmental changes likely to occur from the anthropogenic activity of the entire human population. The figures showing the provincial sets of histograms for the evaluation of the various sub-systems are interesting, but it is unclear whether the increases in computed “harmony” are of any statistical significance. Some measure of scale/units of the heights of the histograms would be desirable. The meaning of figure 10 is quite unclear.
In the short section on regulation of twelve indicators, some of which it was asserted cannot be regulated (such as coal reserves and natural gas production), it is unclear why they cannot be regulated. Are some of the indicators inappropriate? Should coal production rather than reserves be an indicator, as is the case with natural gas? The issue is not the size of the reserves but how and whether they are to be managed. One assumes they will be managed in a transition to a low emissions economy.
The paper gives an interesting array of responses to eight alternative regulatory options, but does not discuss the feasibility of implementing any of them. The paper would benefit from discussion on how the alternative regulatory options might be evaluated and a choice made. The paper does not show any consultation with stakeholders currently involved in the various subsystems.
Whilst this academic paper describes an interesting set of transformations of a variety of parameters involved in water, energy and food subsystems, the authors do recognise the limitations of what they have explored. If any policy decisions have been made based on harmony regulation options, it would be helpful if they were described..
Author Response

(The authors gave the same response as above.)

Round 2
Reviewer 3 Report
The authors have addressed the principal concern regarding lack of clarity for readers on the concept of Harmonious Relationships. They advise that some of the other suggestions made cannot be addressed at this stage and need further research work.
Figure 10 remains incomprehensible. It needs a much more encompassing caption and the coloured diagonal links between water, energy and food and the various indicators need to be darker, and the plethora of links between the indicators and the provinces need to be bolder as neither sets of links are visible on a printed copy and are barely visible on an enlarged screen copy.
On a minor point, the word “Human” on page 2, line 64 should be replaced either by “humanity” or “Humans”.
Author Response
Dear Reviewer,
Thank you again for your comments and suggestions on our manuscript entitled " Analysis and Regulation of the Harmonious Relationship among Water, Energy and Food in nine provinces along the Yellow River " (ID: Water-1615475). Those comments and suggestions are very valuable and helpful for revising and improving our manuscript.
Based on your comments, we have made a second revision. This has helped us greatly to improve our manuscript.
The corrections in the manuscript and responses to you are listed below. We would like to re-submit this revised manuscript to WATER, and hope this manuscript is suitable for the journal now.
We deeply appreciate your hard work and consideration of our manuscript, and look forward to hearing from you soon.
Thank you and best regards.
